# A Novel Method to Screen Strong Constitutive Promoters in *Escherichia coli* and *Serratia marcescens* for Industrial Applications

**DOI:** 10.3390/biology12010071

**Published:** 2022-12-30

**Authors:** Xuewei Pan, Mi Tang, Jiajia You, Yanan Hao, Xian Zhang, Taowei Yang, Zhiming Rao

**Affiliations:** 1Key Laboratory of Industrial Biotechnology of the Ministry of Education, Laboratory of Applied Microorganisms and Metabolic Engineering, School of Biotechnology, Jiangnan University, Wuxi 214122, China; 2School of Food Science and Technology, Jiangnan University, Wuxi 214122, China

**Keywords:** random genomic interruption, constitutive promoter, FACS technology, L-valine, prodigiosin

## Abstract

**Simple Summary:**

With the advancement of synthetic biology and metabolic engineering, regulatory elements applied for the accurate expression of target genes have become more important. Among them, due to their important role in regulating gene expression at the transcription level, a number of homologous or heterologous promoters have been used to improve the yield of target metabolites in different microorganisms. However, the method to isolate strong constitutive promoters in different microorganisms is still limited. Our work describes a novel approach to identify strong constitutive promoters in *Escherichia coli* and *Serratia marcescens*. The identified promoters were further used for fine-tuning gene expression and reprogramming the metabolic flux of L-valine and prodigiosin in *E. coli* and *S. marcescens*, respectively, and finally, the higher-level L-valine synthesis strain and prodigiosin production strain were isolated. The method shown in our study can also be a useful strategy to identify strong constitutive promoters in other bacteria and isolate other effective genetic regulatory elements, such as ribosome binding sites, terminators, and N-terminal coding sequences (NCS), for tuning gene expression in different microorganisms.

**Abstract:**

Promoters serve as the switch of gene transcription, playing an important role in regulating gene expression and metabolites production. However, the approach to screening strong constitutive promoters in microorganisms is still limited. In this study, a novel method was designed to identify strong constitutive promoters in *E. coli* and *S. marcescens* based on random genomic interruption and fluorescence-activated cell sorting (FACS) technology. First, genomes of *E. coli*, *Bacillus subtilis*, and *Corynebacterium glutamicum* were randomly interrupted and inserted into the upstream of reporter gene *gfp* to construct three promoter libraries, and a potential strong constitutive promoter (P_BS_) suitable for *E. coli* was screened via FACS technology. Second, the core promoter sequence (P_BS76_) of the screened promoter was identified by sequence truncation. Third, a promoter library of P_BS76_ was constructed by installing degenerate bases via chemical synthesis for further improving its strength, and the intensity of the produced promoter P_BS76-100_ was 59.56 times higher than that of the promoter P_BBa_J23118_. Subsequently, promoters P_BBa___J23118_, P_BS76_, P_BS76-50_, P_BS76-75_, P_BS76-85_, and P_BS76-100_ with different strengths were applied to enhance the metabolic flux of L-valine synthesis, and the L-valine yield was significantly improved. Finally, a strong constitutive promoter suitable for *S. marcescens* was screened by a similar method and applied to enhance prodigiosin production by 34.81%. Taken together, the construction of a promoter library based on random genomic interruption was effective to screen the strong constitutive promoters for fine-tuning gene expression and reprogramming metabolic flux in various microorganisms.

## 1. Introduction

Microorganisms have been widely used to synthesize numerous chemicals and materials that have previously been produced from fossil resources, especially in the fermentative production of amino acids, antibiofiguretics, enzymes, and biofuels [1,2,3,4]. To achieve product accumulation at the desired levels, efficient genetic tools for controlling the key gene expression of related biosynthetic pathways are indispensable [3,5,6]. Hence, with the demand for the precise quantification of target gene expression in the dynamic and fine-tuned regulation of metabolic flux in synthetic and systems biotechnology, gene expression tools and systems have been rapidly established and developed for mediating gene expression and balancing metabolic flux in various organisms [7,8]. Genetic regulatory elements are of great importance for metabolic engineering and synthetic biological applications, allowing the precise regulation of gene expression at the desired levels [9,10]. At present, genetic regulatory elements for mediating gene expression mainly rely on the transcriptional, post-transcriptional, translational, and protein degradation levels, and include promoters, ribosome binding sites, terminators, small regulatory RNAs, and so on [11,12,13,14,15]. Among them, promoter engineering has emerged as a powerful tool to strongly control gene expression at the level of transcription and has a significant role in tuning the expression of downstream genes; thus, it is widely employed for improving the yield of target metabolites in different microorganisms [16,17].

Generally, transcriptional regulation of key genes is a direct and resultful approach to balance metabolic flux and optimize gene expression [18,19]. To accurately regulate the metabolic flux of the products, it is essential to construct an ingenious method to generate a promoter library covering a wide range of promoters of various intensities, from which promoters of appropriate strength can be screened. In recent years, promoter libraries have been created by promoter modification, omics screening, de novo design, and so on [20,21]. Moreover, constitutive promoters have commonly acted as the regulatory element to realize the regulation of a metabolic network in multiple organisms, such as *E. coli*, *C. glutamicum*, *B. subtilis*, Streptomyces, and some non-model strains [22,23,24,25]. In *Halomonas bluephagenesis*, the promoter (Pporin) of the strongest-expressed protein porin was used to construct a library with a wide range of intensities. The promoter library was constructed via saturation mutagenesis in the core region of the promoter, covering the relative transcriptional strengths from 40 to 14,000, and could work in both *E. coli* and *H. bluephagenesis* for the expression of heterologous genes [26]. Due to the deficiency of available promoters in Burkholderiales, an approach was carried out to screen strong constitutive promoters based on transcriptome sequencing. Thirty-seven promoters identified from the omics data were then cloned and characterized with a firefly luciferase reporter and applied to drive the 56 kb epothilone BGC and 23 kb rhizomide BGC for the efficient production of epothilone and rhizomide [27]. In *Bacillus licheniformis* DW2, a gradient strength promoter library was obtained by coupling the bacitracin synthetase gene cluster promoter PbacA with various 5′-UTRs. The screened promoters with strong, middle, and weakest intensities were then successfully used in protein expression and metabolic pathway optimization [28]. On the basis of the endogenous tandem promoter Pldh, a promoter library was developed to fine-tune the expression of the gene *pyc* to enhance malic acid biosynthesis in *Bacillus coagulans* [29]. Moreover, a flow cytometry-based quantitative method was constructed for gene expression, and 200 native or synthetic promoters were identified in a high-throughput platform in Streptomyces [17]. In addition, to realize the dynamic regulation of gene expression, exogenous inducers are required to control the timing of gene expression, but the inducers are usually expensive and unstable, making theme unsuitable for industrial application. Alternatively, self-regulated promoters are workable to dynamically modulate gene expression without the addition of any triggers, and have thus been widely used in balancing cell growth and product synthesis [30]. Above all, although promoters suitable for target gene expression could be successfully screened from the designed promoter libraries, the methods to screen strong constitutive promoters in different microorganisms are still limited, and most were obtained by random mutations on the reported promoters or by identification from the omics data [31,32,33,34,35,36]. Hence, more effort should be made to design a novel method to identify strong constitutive promoters in different bacteria. 

In this study, an approach based on random genomic interruption and FACS technology was designed to isolate strong constitutive promoters in *E. coli* and *S. marcescens*. Green fluorescent protein (GFP) was used as a reporter gene, and FACS technology was applied to sort high-intensity promoters. After identification, characterization, and modification of the strong constitutive promoters screened from these libraries, the final regulatory elements were applied to enhance the metabolic flux in the biosynthesis of L-valine and prodigiosin to confirm the strength of the identified promoters. The main workflow involved: (1) Randomly interrupting the genomes of *E. coli*, *B. subtilis*, and *C. glutamicum* to construct three different promoter libraries, respectively; (2) Characterizing of the identified strong constitutive promoter; (3) Constructing a gradient strength promoter library based on the modification of promoter P_BS76_; (4) Applying the regulatory element to enhance the expression of *ilvCDE* genes for L-valine overproduction in *E. coli* W3110; (5) Screening for strong constitutive promoters applicable for *S. marcescens* and utilizing the promoter to optimize the metabolic flux of prodigiosin synthesis in *S. marcescens* JNB5-1. Therefore, the method shown in our study can also be a useful strategy to identify strong constitutive promoters in other bacteria. 

## 2. Materials and Methods

### 2.1. Strains, Plasmids, and Cultivation

Strains and plasmids used in this study are listed in Appendix A. *E. coli* JM109 was used as the host for vector construction. *E. coli* MG1655, *B. subtilis* 168, and *C. glutamicum* ATCC13032 were employed as the targets for random genomic interruption manipulation. *S. marcescens* JNB5-1 was used to screen strong constitutive promoters for application in prodigiosin synthesis. Among them, *E. coli* and *B. subtilis* were grown on lysogeny broth (LB) medium (yeast extract 5 g/L, tryptone 10 g/L, NaCl 10 g/L) at 37 °C for 8–12 h. *C. glutamicum* was cultured in brain heart infusion (BHI) medium at 30 °C for 16–24 h. *S. marcescens* was cultivated overnight in LB medium at 30 °C. When necessary, 100 µg/mL ampicillin was added into the medium for selections.

### 2.2. Design and Construction of Random Library to Isolate Strong Constitutive Promoters Based on Random Genomic Interruption

To construct random libraries to isolate strong constitutive promoters in *E. coli* and *S. marcescens*, genome interruption was carried out. First, genomes of *E. coli* MG1655, *B. subtilis* 168, *C. glutamicum* ATCC13032, and *S. marcescens* JNB5-1 were extracted using TIANamp Bacteria DNA Kit. Here, two approaches (ultrasound and enzyme digestion) were performed to randomly interrupt genomes. In this study, genomes of *E. coli* MG1655 and *S. marcescens* JNB5-1 were interrupted by ultrasonic. For genome fragmentation using an ultrasonic crusher, the output power was set at 20%, the ultrasonic time was 1 s, and the ultrasonic interval was set at 3 s. Ultrasound was performed a total of 30 times to break the genomes into random fragments ranging from 100 to 250 bp. Then, Klenow Fragment (Takara, Beijing, China) was used to repair and smooth the 5′ protruding end of the double-stranded DNA. Specifically, a 20 µL reaction system is described below: Template DNA 25 ng, 10 × Klenow Fragment buffer 2.5 µL, dNTP 2.5 µL, Klenow Fragment 1 µL, and ddH_2_O to 20 µL. After reaction at 37 °C for 3 h, the solution was then incubated at 65 °C for 5 min. Finally, the treated mixed fragment was inserted into the upstream of the report gene *gfp*, and then cloned into the vector pUC19 and pUCP18, to form the plasmid libraries pUC19-P_EC_-*gfp* and pUCP18-P_SM_-*gfp*, respectively. 

Genomes of *B. subtilis* 168 and *C. glutamicum* ATCC13032 were interrupted by enzyme digestion. *Mbo* I (Takara, Beijing, China) was used to achieve the interruption of *B. subtilis* 168 and *C. glutamicum* ATCC13032 genomes. The restriction site identified by *Mbo* I was GATC, and linear DNA fragments of different sizes would be obtained in the places with this restriction site in the genome. In the 20 μL reaction solution, the amount of *Mbo* I added was 1 μL, the amount of 10× K buffer added was 2 μL, DNA fragments added were less than 1 μg, and the rest was supplemented with ddH_2_O. Then, the reaction solution was incubated at 37 °C for 20 min. The linearized fragments obtained by enzyme digestion need dephosphorylation before being cloned into the vector. Thus, alkaline phosphatase (Takara, Beijing, China) was used to remove the phosphate group at the 5′ end of DNA fragments. DNA fragments (1–10 pmol), 5 μL of alkaline phosphatase, and 10× SAP buffer were added to the reaction system, which was made up to 50 μL with sterile water. First, the solution was reacted at 37 °C for 15–30 min, and then it was incubated at 65 °C for 15 min to inactivate it. At this point, 2.5 μL of 3 M NaCl and precooled absolute ethanol were added, and the solution was placed at −20 °C for 1 h. The solution was centrifuged and washed by 200 μL of precooled 70% ethanol, and finally, it was dried and dissolved with TE buffer. Similarly, random DNA fragments of *B. subtilis* 168 and *C. glutamicum* ATCC13032 genomes were inserted into the plasmid pUC19-*gfp* to obtained plasmids libraries pUC19-P_BS_-*gfp* and pUC19-P_CG_-*gfp*, respectively. Then, plasmids pUC19-P_EC_-*gfp*, pUC19-P_BS_-*gfp*, and pUC19-P_CG_-*gfp* were transformed into *E. coli* JM109 to isolate strong constitutive promoters in *E. coli*, as the expression of fluorescent GFP is driven by the random genomic regions. pUCP18-P_SM_-*gfp* was transformed into *S. marcescens* JNB5-1 to isolate strong constitutive promoters in *S. marcescens*. The transformants incubated on the LB plates were collected and washed with PBS buffer three times, and then the OD_600_ was diluted to about 0.3, followed by FACS sorting. The recombinant strains *E. coli* and *S. marcescens* with higher fluorescence were coated on the plates after FACS sorting for overnight culture. All the colonies grown on the plates were further inoculated into 96-deep-well plates and cultured at 37 °C. GFP fluorescence and optical density (OD_600_) were detected using the microplate reader after 10 h of incubation. 

### 2.3. Construction of the Promoter P_BS76_ Library

The truncated promoter P_BS76_ was applied to construct four random mutation libraries. Thus, P_BS76_ was ligated with the reporter gene *gfp* and cloned into vector pUC19 to construct the basic plasmid for subsequent library construction. Four mutation libraries were constructed using the basic plasmid pUC19-P_BS76_-*gfp* as a template through PCR with primers containing degenerate bases. The PCR ligation product was then transformed into the competent cells of *E. coli* JM109. All the colonies on the LB plates were collected and subjected to plasmid extraction, and then the plasmid libraries were electroporated into *E. coli* MG1655. The strains incubated on the plates were collected and washed with PBS buffer three times, and then the OD_600_ was diluted to about 0.3, followed by FACS sorting. Colonies with higher fluorescence intensity than that of the control were selected and spread on the LB plates containing ampicillin (100 µg/mL). The recombinant *E. coli* strains containing the fluorescence plasmids grown on the LB agar plates were collected and cultured in 96-deep-well plates at 37 °C for the second-round screening with enhanced GFP fluorescence. After overnight incubation, GFP fluorescence and optical density (OD_600_) were measured using the microplate reader. 

### 2.4. Construction of Recombinant Strains

The vector used for promoter screening and validation was pUC19, and the reporter gene *gfp* was used to characterize the promoter strength. The construction procedure of *gfp* expression plasmid pUC19-P_BBa_23118_-*gfp* served as an example. In brief, promoter P_BBa_23118_ and gene *gfp* were amplified by corresponding primers (Appendix A) and fused by overlap extension PCR. Then, the fused fragment was inserted into the linearized vector pUC19 using ClonExpress One Step Cloning Kit (Vazyme Biotech Co., Nanjing, China), resulting in plasmid pUC19-P_BBa_23118_-*gfp*. The recombinant plasmid was then transformed into *E. coli* JM109 and selected on an LB agar plate containing ampicillin (100 µg/mL). Similarly, the other GFP expression vectors under the control of different promoters were constructed using this approach. As for the gene expression plasmids constructed based on expression vectors pTrc99a and pUCP18 for the overexpression of genes *ilvCDE* and *pigFN*, respectively, the genes *ilvCDE* and *pigFN* were amplified and purified, and then the purified PCR products were ligated with the linearized plasmids pTrc99a and pUCP18 through Gibson assembly (ClonExpress^®^ II One Step Cloning Kit, Vazyme Bio Inc., Nanjing, China). The recombinant plasmids were transformed into strains *E. coli* W3110 and *S. marcescens* JNB5-1 by electroporation, and selected on LB agar plate containing ampicillin (100 µg/mL). 

### 2.5. Fluorescence Assays

The recombinant *E. coli* strains that contained the fluorescence plasmids or the plasmid libraries were inoculated in LB medium and cultured at 37 °C, 200 rpm for 10 h in the plate shaker. The cultures were diluted with 0.1 M phosphate-buffered saline (PBS, pH 7.4) to guarantee an OD_600_ value of about 0.5 before fluorescence detection. Then, 200 µL of each culture was transferred into the 96-deep-well black plates, and the whole plate was measured using the microplate reader (Cytation 3, BioTek Instruments, Inc., Winooski, VT, USA) under an excitation wavelength of 490 nm and an emission wavelength of 530 nm at 25 °C. Fluorescence intensity was determined by dividing the fluorescence by OD_600_. The background fluorescence value of strains without fluorescence expression (FP_bg_) and the background OD_600_ of the medium (OD_bg_) were corrected, and the following Equation (1) was used to calculated the relative fluorescence intensities.
(1)$$\left(\frac{\mathrm{FP}}{\mathrm{OD}}\right)=\left(\frac{\mathrm{FP}-{\mathrm{FP}}_{\mathrm{bg}}}{\mathrm{OD}-{\mathrm{OD}}_{\mathrm{bg}}}\right)$$

### 2.6. Cultivation in Shake Flasks

The engineered strains modified for L-valine production were first activated on LB agar plates, then inoculated into 30 mL seed medium in the 500 mL shake flask and cultivated for 10 h at 37 °C and 230 rpm. The seed medium contained 20 g/L glucose, 10 g/L yeast extract, 5 g/L peptone, 1.2 g/L KH_2_PO_4_, 1.2 g/L MgSO_4_·7H_2_O, 10 mg/L FeSO_4_·7H_2_O, 10 mg/L MnSO_4_·H_2_O, 1.3 mg/L VB_1_, and 0.3 mg/L V_H_. Then, 5 mL seed culture was transferred into 25 mL fermentation medium in a 500 mL shake flask and cultivated for 24 h at 37 °C and 230 rpm. The fermentation medium contained 20 g/L glucose, 2 g/L yeast extract, 4 g/L peptone, 1 g/L NaCl, 2 g/L KH_2_PO_4_, 0.7 g/L MgSO_4_·7H_2_O, 0.1 g/L FeSO_4_·7H_2_O, 0.1 g/L MnSO_4_·H_2_O, 0.8 mg/L VB_1_, and 0.2 mg/L V_H_. During the fermentation process, pH was maintained at 7.0 by adding NH_4_OH with phenol red as an indicator and adding 60% glucose solution when the glucose concentration was low. 

For prodigiosin production, the engineered strains were grown overnight at 30 °C and 200 rpm in a rotary shaker, then transferred into the 250 mL shake flask containing 30 mL fermentation medium for 72 h cultivation at 30 °C. The fermentation medium contained 20 g/L sucrose, 15 g/L beef extract, 10 g/L CaCl_2_, 7.5 g/L L-proline, 0.2 g/L MgSO_4_·7H_2_O, and 6 mg/L FeSO_4_·7H_2_O. 

### 2.7. Analytical Methods

Cell growth was measured by the UV spectrophotometer at OD_600_. The L-valine production was determined by HPLC using acetonitrile/water (10⁚90 *v*/*v*) as the mobile phase. The flow rate was 1 mL/min, and detection wavelength was set as 278 nm. Glucose concentration was detected by using a glucose biosensor (SBA-40C, Shandong, China). Prodigiosin was measured as previously reported with minor modification [37]. The fermentation medium was dissolved in acidic ethanol (pH 3.0) and placed for 8 h until fully dissolved. Then, the samples were centrifuged at 10,000 rpm for 10 min. Finally, the supernatant was taken for determination at the absorbance of A_535_. Fluorescence characterization with cytometry was evaluated with a BD FACS AriaII cell sorter (BD FACSDiscover™ S8, Becton, Dickinson and Company, Franklin Lakes, NJ, USA). The collected cells were diluted into cold PBS before detection, loaded, and run at the rate of 0.5 µL/s. A total of 10,000 samples were captured for subsequent sorting.

All the experiments were carried out in triplicate, and all the data are expressed as the mean ± standard deviation. One-way ANOVA was used for comparing statistical difference between the groups of experimental data.

## 3. Results

### 3.1. Screening of Potential Strong Constitutive Promoters Based on a Novel Approach of Random Genomic Interruption and FACS Technology

In microorganisms, protein expression level is affected by many factors, such as transcription initiation, transcription termination, translation regulation, and RNA stability, among which transcription initiation plays an essential role. Promoters, as the switch to control transcription initiation, play a key role in regulating the output level of genes. Therefore, promoters are often modified to achieve the appropriate expression levels of the target gene. Currently, screening promoters of appropriate strength by constructing various promoter libraries is the most common approach. However, the methods for isolating strong constitutive promoters in microorganisms are still limited, and are mainly based on promoter modification and omics screening. In this study, a novel approach to screen strong constitutive promoters based on random genomic disruption and FACS technology was designed (Figure 1A). Genomes of *E. coli* MG1655, *B. subtilis* 168, and *C. glutamicum* ATCC13032 were extracted, randomly interrupted, and inserted into the upstream of the reporter gene *gfp* for further construction of promoter libraries. Specifically, genomes of *B. subtilis* 168 and *C. glutamicum* ATCC13032 were first digested using the enzyme *Mob* I, and the reaction solution was reacted at 37 °C for 20 min. As shown in Figure 1B, the broken genomic fragments were mainly concentrated around 100–250 bp, shown in the lane 2 and 3. The genome of *E. coli* MG1655 was fragmented by ultrasound, and the power was set to 20%. After 30 s of fragmentation, the band presented in lane 1 was obtained. The resulting fragments from the three sources were inserted into vector pUC19, upstream of the reporter gene *gfp*, respectively. At this point, as the expression of fluorescent GFP was driven by the random genomic regions, three random promoter libraries were generated (Library1-EC, Library2-BS, Library3-CG). Then, all transformants grown on the LB plates were collected and subjected to FACS sorting. *E. coli*/pUC19-P_BBa_J23118_-*gfp* was used as the control strain to screen mutants with significantly higher fluorescence intensity. In Library1-EC, the proportion of strains with significantly higher fluorescence value accounted for 0.88% of the total library, while in Library2-BS and Library3-CG, the proportion of strains with significantly higher fluorescence value accounted for 1.52% and 0.1% of the total library, respectively (Figure 1C). The sorted strains were then coated on the LB plates and inoculated in 96-deep-well plates for rescreening. Obviously, the fluorescence values of about 90% of the mutants were higher than that of the control strain. Moreover, in each library, there was a mutant with significantly higher fluorescence intensity than that of the other mutants (Figure 1D). As shown in Figure 1E, on the LB plates, the color of the three mutants with the stronger fluorescence intensities was significantly greener than that of the control. Fluorescence microscopy also showed that the fluorescence intensity of the mutant cells was significantly higher than that of control strain. Finally, the fluorescence values of the three mutants that contained the *gfp* gene under the control of promoters from different sources were compared. The results showed that the fluorescence intensity of the mutant containing the *gfp* gene under the control of the promoter from *B. subtilis* was significantly higher than that of the other two mutants, and about 19.7 times higher than that of the control strain (Figure 1F). Thus, the screened random genomic region that contained sequences of the strong constitutive promoter from *B. subtilis* 168 was selected for further study. In conclusion, the construction of the promoter library based on random genomic interruption and FACS technology could be effectively applied in the screening of strong constitutive promoters in *E. coli*. 

### 3.2. Characterization of the Identified Strong Constitutive Promoter from B. subtilis 168

In order to further analyze and identify the random sequence obtained from *B. subtilis* 168, the plasmid of the mutant was sequenced. The Results show that the random sequence upstream of the reporter gene *gfp* was the 136 bp sequence upstream of the transcription factor AcoR, which regulates acetoin synthesis in *B. subtilis*. Subsequently, the 136 bp sequence was shortened from both ends to identify the sequences of the promoter region of the *acoR* gene. As shown in Figure 2A, the sequence was truncated from the 5′ end to the 3′ end, with each truncation being 20 bp, until the sequence was truncated to 16 bp, and a total of six truncated DNA fragments (P1–P6) were obtained. Similarly, the sequence was truncated from the 3′ end to the 5′ end by 20 bp each time, and six other DNA fragments (P7–P12) were also obtained. The truncated DNA fragments were ligated before the reporter gene *gfp* by PCR amplification, cloned into the vector pUC19, and then transformed into *E. coli* JM109 for verification. Data showed that as the sequence was truncated from 5′ end to 3′ end to 116 bp, 96 bp, and 76 bp, the fluorescence intensity was basically consistent with that of the original DNA fragment. However, when the sequence truncated to 56 bp, 36 bp, and 16 bp, the expression level of GFP decreased significantly. In addition, when the sequence was truncated from 3′ end to 5′ end, only DNA fragment P7 had the same fluorescence intensity as that of the original DNA fragment. After that, the other truncated sequences showed a significantly lower fluorescence value than that of the original sequence. (Figure 2B). Subsequently, the fluorescence intensity of DNA fragments P1–P6 was further analyzed by FACS analysis to verify the above experimental results. The fluorescence intensity of P1, P2, and P3 was consistent with that of the original DNA fragment, while the fluorescence value of P4, P5, and P6 was significantly weaker (Figure 2C). These results indicated that the sequences located between positions −82 and −7 relative to the TIS of the *acoR* gene contained the identified strong constitutive promoter (here we named P_BS76_). These data are consistent with the result of N O Ali et al. [38]. The identified P_BS76_ was used as the subsequent research object. Above all, sequence truncation is an effective way to identify the promoter regions of the screened sequence. 

### 3.3. Construction of a Gradient Promoter Library via Modifying Promoter P_BS76_

To achieve high yields of target metabolites, appropriate promoters should be screened and used for fine-tuning gene expression and reprogramming the metabolic flux of target products. Theoretically, the strength of the promoter is determined by adjacent sequences from the −35 region to the transcription start site. Hence, to better regulate the expression of key enzymes and enhance the metabolic flux of target product synthesis, the screened strong constitutive promoter P_BS76_ was further modified by the construction of promoter libraries. The workflow of the construction and screening of the promoter random mutagenesis library is shown in Figure 3A. As shown in Figure 3B, four promoter P_BS76_ mutation libraries were designed and constructed with sequences of −35 region (Library 1), spacer sequences between −35 and −10 regions (Library 2), sequences of −10 region (Library 3), and sequences between −10 region and transcription start site (Library 4) modified by the introduction of random base “N”, respectively. Firstly, random base “N” was introduced into the four regions of P_BS76_ promoter in plasmid pUC19-P_BS76_-*gfp* by inverse PCR, and the resulting plasmids containing mutation libraries were transferred into *E. coli* JM109, respectively. Then, FACS sorting technology was applied to screen the mutant strains with a higher fluorescence intensity than that of the control. The results showed that the portion with a higher fluorescence intensity in Library 1, Library 2, Library 3, and Library 4 accounted for 0.9%, 1.6%, 1.1% and 2.1% of the total library, respectively. As shown in Figure 3C, the fluorescence intensity of Library 1 and Library 3 mainly fluctuated in the range of 0–10^4^ (a.u.), and the activity of most promoters was concentrated between 0 and 10^3^ (a.u.). The proportion of mutant strains with a high fluorescence value in Library 1 and Library 3 was relatively small. The fluorescence range of mutants in Library 2 and Library 4 was mainly in the range of 10^3^–10^5^ (a.u.), which was higher than that in Library 1 and Library 3. Thus, mutants with a significantly higher fluorescence intensity than that of the control could be easier to screen in Library 2 and Library 4. The mutants with a higher fluorescence intensity sorted from Library 2 and Library 4 were coated on LB plates for overnight cultivation. The grown colonies were randomly selected and inoculated in 96-deep-well plates, and fluorescence intensity was measured after 8 h of incubation. The results indicated that a mutant (P_BS76-variant_) with a significantly higher fluorescence intensity than that of others was screened using promoter P_BS76_ as the control (Figure 3D). The intensity of P_BS76-variant_ was 59.56 times higher than that of promoter P_BBa_J23118_ (Figure 3E). In total, four promoter libraries were constructed to obtain a series of promoters with a gradient of intensity, among which the strongest one was 59.56 times higher than that of promoter P_BBa_J23118,_ and 3.03 times higher than that of promoter P_BS76_. In the following experiments, to further confirm the strength of the promoters identified in our study, six promoters, P_BBa_J23118_, P_BS76_, P_BS76-50_, P_BS76-75_, P_BS76-85_, and P_BS76-100_, with different strengths were selected for the regulation of expression levels of key genes and metabolic flux in L-valine synthesis.

### 3.4. Application of the Identified Regulatory Elements for L-Valine Overproduction in E. coli W3110

To obtain the higher-level titer of target products, proper metabolic engineering should be carried out for the target metabolite. Here, to further validate the promoter activities identified in our study, L-valine, a branched-chain amino acid, which is widely used in nutrient supplements and the pharmaceutical industry, served as the target metabolite [39,40]. Based on the synthetic pathway of L-valine, three key enzymes (acetohydroxy acid isomeroreductase, encoding by *ilvC*; branched-chain amino acid aminotransferase, encoding by *ilvE*; dihydroxy acid dehydratase, encoding by *ilvD*) involved in L-valine biosynthesis were subjected to manipulate for better production of L-valine (Figure 4A). A L-valine-producing strain (Val01) obtained by ARTP mutagenesis of *E. coli* W3110 was used as the starting strain [41]. The L-valine production of Val01 could reach 2.90 g/L after 24 h fermentation in shake flask. Promoters P_BBa_J23118_, P_BS76_, P_BS76-50_, P_BS76-75_, P_BS76-85_, and P_BS76-100_ with different strength were applied to enhance the metabolic flux of L-valine synthesis. Specifically, the key *ilvCDE* genes were cloned into the vector pTrc99a under the control of promoters P_BBa_J23118_, P_BS76_, P_BS76-50_, P_BS76-75_, P_BS76-85_, and P_BS76-100_, respectively, and then transformed into *E. coli* JM109. The transformants were selected and verified by colony PCR and sequencing. The successful transformants were transformed into strain Val01, resulting in recombinant strains Val02, Val03, Val04, Val05, Val06, and Val07 (Figure 4B). Subsequently, the recombinant strains were subjected to shake flask fermentation for 24 h to compare their ability to produce L-valine. Based on our results, the cell biomass of strains Val02, Val03, Val04, Val05, and Val06 was not different from that of strain Val01 after 24 h shake-flask fermentation, while the OD_600_ of strain Val07 only reached 28. Obviously, the intensity of promoters was positively correlated with the L-valine yield produced by these seven strains. Among them, the L-valine yield produced by Val07 reached 7.92 g/L, while amounts of 3.98, 5.12, 6.55, 7.03, and 7.53 g/L L-valine were attained by strains Val02, Val03, Val04, Val05, and Val06, respectively (Figure 4C). Strain Val07 owned the highest L-valine yield, 173.1% higher than that of strain Val01. The strength of promoter P_BS76-100_ was too strong, which caused the disorder of intracellular metabolism and the burden on cell growth, so the cell growth was severely inhibited. However, due to the enhanced metabolic flux of L-valine synthesis driven by promoter P_BS76-100_, there was an improvement in L-valine yield. Taken together, these results further confirmed that promoters P_BS76_, P_BS76-50_, P_BS76-75_, P_BS76-85,_ and P_BS76-100_ are strong promoters compared to promoter P_BBa_J23118_ and could be used to improve the production of target products. 

### 3.5. Screening of Strong Constitutive Promoter Applicable for S. marcescens by Random Genomic Disruption and FACS Technology

Prodigiosin is a red pigment with great economic value and of widely promising application due to its immunosuppressive and anticancer activities [37]. *S. marcescens* is the main producing strain for prodigiosin synthesis. Research of the model organism for prodigiosin synthesis has mainly focused on transcriptional regulation, the function of a two-component system and a quorum sensing system [42,43,44]. However, due to the unclear genetic background and the lack of gene-editing approaches of *S. marcescens*, the improvement in prodigiosin production has been severely limited. Moreover, the shortage of an effective synthetic biology toolbox to precisely regulate gene expression has impeded its development seriously. Here, to further confirm the general applicability of the method conducted in our study to screen strong constitutive promoters in different bacteria, a gradient strength promoter library was constructed via random genomic disruption for screening strong constitutive promoters in *S. marcescens* as mentioned above (Figure 5A). First, the genome of *S. marcescens* JNB5-1 was extracted and interrupted by ultrasonic. The output power of the ultrasonic crusher was set at 20%, the ultrasonic time was 1 s, and the ultrasonic interval set at 3 s. A total of 30 repeats of ultrasound were achieved to break the genome into random fragments ranging from 100 to 300 bp (Figure 5B). The 5′ end of the fragments was repaired and smoothed by the Klenow Fragment. Then, the random fragments were inserted into the upstream of the reporter gene *gfp* and cloned into the vector pUCP18, resulting in the plasmid library pUCP18-P_SM_-*gfp*. The constructed plasmid was then transformed into *S. marcescens* JNB5-1, and the transformants grown on the LB plates were collected for further FACS sorting. *S. marcescens* JNB5-1 containing plasmid pUCP18-P_pig_-*gfp* was used as the control, which was used to express the gene *gfp* under the control of the native promoter P_pig_ of prodigiosin synthesis gene cluster. Mutants with a higher GFP fluorescence than that of the control were collected for further analysis. The results showed that the fluorescence intensity varied over a 1000-fold range from 10^2^ to 10^5^ (a.u.), and the activities of most promoters were mainly concentrated between 10^3^ and 10^4^ (a.u.) (Figure 5C). Moreover, the sorted strains were then coated on LB plates, and the colonies were inoculated into 96-deep-well plates for the second round of screening. Among them, the GFP expression intensity of 95% mutants was higher than that of the control (Figure 5D). The fluorescence intensity of one of the mutants containing the *gfp* gene under the control of the promoter P_SM_ (the promoter of the operon *rpsQ*-*rpsJ*) was significantly higher than that of the others. Therefore, the random sequence of this mutant was selected for further research.

Nowadays, prodigiosin attracts much attention due to its multiple biological activities. In *S. marcescens*, the synthesis of prodigiosin is achieved by the *pig* gene cluster. 2-methyl-3-n-amyl-pyrrole (MAP) and 4-methoxy-2,2′-bipyrrole-5-carbaldehyde (MBC) are two major precursors and are condensed into prodigiosin via the PEP-utilizing enzyme. Importantly, O-methyl transferase (encoded by *pigF*) and oxidoreductase (encoded by *pigN*) involved in catalyzing 4-hydroxy-2,2′-bipyrrole-5-carbaldehyde (HBC) to the formation of 4-methoxy-2,2′-bipyrrole-5-carbaldehyde (MBC) have been confirmed to play the most significant role in prodigiosin synthesis among the *pig* gene cluster in our previous study (Figure 5E). In addition, substituting the promoter of a gene cluster with strong constitutive promoters has been proven as a feasible strategy for metabolite production. Here, to study the application of the strong constitutive promoter P_SM_ screened from *S. marcescens* in the optimization of the prodigiosin metabolic pathway, the gene expression of *pigFN* was regulated by replacing promoters with those of different strengths. The native promoter P_pig_ of the *pig* gene cluster, an endogenous promoter P_RplJ_ from *S. marcescens* found using RNA-Seq [37], and the novel constitutive promoter P_SM_ that was screened based on random genomic interruption were selected to mediate the expression of *pigFN*. Thus, the expression vectors of *pigFN* genes mediated by promoters P_pig_, P_RplJ_, and P_SM_ were then transferred into *S. marcescens* JNB5-1, resulting in strains SM01, SM02, and SM03. Then, these strains were cultivated in fermentation medium for 72 h. As shown in Figure 5F, after 72 h of fermentation, the prodigiosin production of the wild-type JNB5-1 was 6.32 g/L. The yield of prodigiosin in strains SM01 and SM02 reached 7.50 g/L and 7.91 g/L, respectively. The maximum prodigiosin titer of strain SM03 reached 8.52 g/L, which was 34.81% higher than that produced by the wild-type strain. Meanwhile, the activity of promoter P_SM_ was stronger than that of promoter P_RplJ_ in the *S. marcescens* expression system. Altogether, these results suggested that the method based on random genomic disruption and FACS technology could also be a valuable approach to screen strong constitutive promoters in *S. marcescens*.

## 4. Discussion

The transcription of DNA to RNA is the first step in protein expression, and promoter, RNA polymerase, and sigma factor are typically involved in this process. Among them, promoters directly affect the transcription rate of protein, and hence they are common targets used for engineering the biosynthesis of natural and non-natural products in model or non-model strains [45]. To improve L-proline production in *C. glutamicum*, tailored promoter libraries were used to fine-tuned the expression levels of the target *gdh*, *pyc*, and *proB* genes, and finally, L-proline production was significantly increased [46]. To balance the metabolic flux of the naringenin biosynthesis pathway in *E. coli*, constitutive promoters with a gradient of intensity were randomly selected to control the expression levels of the pathway genes of naringenin. After screening more than 1200 candidates via the ultraviolet spectrophotometry–fluorescence spectrophotometry high-throughput method, the metabolic flux of the naringenin synthetic pathway was appropriately balanced, and finally, the naringenin was significantly improved [47]. Song et al. screened promoters from *B. subtilis* in various conditions, and a strong promoter, PtrnQ, in comparison to P43, was isolated and used to elevate the final production of both cytoplasmic BgaB and secreted protein α-amylase [36]. Promoter engineering has been also used to improve the production of 3-aminopropionic acid [48] and myo-inositol in *E. coli* [49], pullulanase [31], methyl parathion hydrolase, and chlorothalonil hydrolytic dehalogenase in *B. subtilis* [50], L-lysine in *C. glutamicum* [51], medium-chain-length polyhydroxyalkanoates in *Pseudomonas mendocina* [32], co-enzyme Q_10_ in *Rhodobacter sphaeroides* [33], polycyclic tetramate macrolactam in *Streptomyces albus* [34], co-enzyme A in *Corynebacterium ammoniagenes* [35], H_2_ in *Thermococcus onnurineus* [52], and so on. However, although promoters suitable for target gene expression and products production could be successfully screened from random mutation on the reported promoters or identified from the omics data, the current methods to screen strong constitutive promoters in different microorganisms are limited. In this study, a novel approach based on random genomic interruption was designed for the construction of promoter libraries, and strong constitutive promoters suitable for *E. coli* and *S. marcescens* were obtained by using FACS technology to sort high-intensity promoter sequences (Figure 1 and Figure 5). Further, to confirm the strength of the identified promoters, the promoters obtained in our study were applied to enhance the expression of *ilvCDE* genes in *E. coli* W3110 to enhance L-valine production, and to increase expression of the *pigFN* genes in *S. marcescens* JNB5-1 to improve prodigiosin synthesis. As shown in Figure 4 and Figure 5, the L-valine production in strain Val07 and prodigiosin synthesis in strain SM03 were significantly increased compared to *E. coli* Val01 and *S. marcescens* JNB5-1 (Figure 4 and Figure 5). As far as we know, our study is the first to identify strong constitutive promoters through the method of random genomic interruption and FACS technology.

L-valine, an essential amino acid, is one of the three branched-chain amino acids (BCAAs), and studies on L-valine have shown that it is widely used in many industrial fields, such as pharmaceuticals, cosmetics, food, and feed [53]. Moreover, with the growing world market for L-valine, the microbial cell factory for the efficient synthesis of L-valine has become of increasing interest. *C. glutamicum* is the most commonly used industrial microorganism for producing L-valine. Due to the advantages of clear genetic background and easy genetic manipulation, *E. coli* has become another important host for the synthesis of L-valine in recent years [54,55]. However, possibly due to the more complex regulatory mechanism for L-valine biosynthesis in *E. coli*, there are fewer reports of L-valine-producing *E. coli* strains than there are of L-valine-producing *C. glutamicum* strains. Park et al. developed an engineered L-valine production strain through systematic metabolic engineering of *E. coli* W, and finally, a high-level production of L-valine (60.7 g/L) with a yield of 0.22 g/g glucose was obtained [54]. Considering that cofactor balance is another important factor required for L-valine biosynthesis, Savrasova et al. constructed an *E. coli* MG1655-based L-valine-producing strain by replacing the native NADPH-dependent aminotransferase with a heterologous NADH-dependent leucine dehydrogenase [56]. By systematic metabolic engineering, Hao et al. constructed a chromosomally engineered L-valine-producing strain, and the final strain could produce 84 g/L L-valine with a yield and productivity of 0.41 g/g glucose and 2.33 g/L/h, respectively, in a 5 L bioreactor through a two-stage fed-batch fermentation [57]. In our previous study, we constructed a high-level L-valine production *E. coli* strain (92 g/L) using multimodular engineering [41]. In this study, to further confirm the strength of the promoters identified in our study, promoters P_BBa_J23118_, P_BS76_ (identified by the method based on random genomic interruption and FACS technology), and four promoters, P_BS76-50_ (weakest), P_BS76-75_ (middle), P_BS76-85_ (strong), and P_BS76-100_ (strongest), from our isolated P_BS76_ promoter library were applied to increase the expression levels of *ilvCDE* genes, and strains Val02, Val03, Val04, Val05, Val06, and Val07 were obtained, respectively. The results of shake flask fermentation showed that the L-valine production in strains Val02, Val03, Val04, Val05, Val06, and Val07 was significantly enhanced compared to the original strain Val01 (Figure 4). These results further support that the method based on random genomic interruption and FACS technology is a very effective method to screen strong constitutive promoters in different bacteria.

Prodigiosin (PG), a red linear tripyrrole pigment, is the most prominent member of the prodiginine family and is produced by *S. marcescens*, *Serratia rubidaea*, *Streptomyces coelicolor*, *Streptomyces griseoviridis*, *Serratia nematodiphila*, and so on [58]. Among them, *S. marcescens* is the most widely studied prodigiosin-producing strain. Studies on prodigiosin have shown that it has important antimicrobial, anticancer, and immunosuppressive properties, and hence prodigiosin has received widespread attention in the last few decades [59]. Due to being economical and more environmentally friendly, the biotechnological production of prodigiosin by *S. marcescens* has recently attracted a great deal of interest. However, the high-efficiency production of prodigiosin by *S. marcescens* for commercial purposes is still a challenge. The prodigiosin biosynthesis pathway in *S. marcescens* is encoded by the *pigABCDEFGHIJKLMN* genes, a total of 14 genes, which are transcribed as a polycistronic mRNA from a promoter upstream of the *pigA* gene. Among them, the genes *pigB*, *pigD*, and *pigE* are involved in the biosynthesis of 2-methyl-3-n-amyl-pyrrole (MAP), while the genes *pigA*, *pigF*, *pigG*, *pigH*, *pigI*, *pigJ*, *pigK*, *pigL*, *pigM*, and *pigN* use L-proline as the substrate to synthesize 4-methoxy-2,2′-bipyrrole-5-carbaldehyde (MBC). Finally, the terminal condensing enzyme PigC encoding by the *pigC* gene condenses MAP and MBC to prodigiosin [58]. In our previous study, via transcriptomics and proteomics, we identified that the genes *pigN* and *pigF* probably play the most important role in prodigiosin biosynthesis in the *pig* gene cluster [60]. In addition, through the introduction of a polynucleotide fragment into the *pigN* 3′ untranslated region and disulfide bonds into the O-methyl transferase (PigF), the prodigiosin production in strain *S. marcescens* JNB5-1 was significantly enhanced [61]. In this study, to verify the universality of the method of screening strong constitutive promoters based on the random genomic interruption and FACS technology, a strong constitutive promoter P_SM_ was firstly isolated based on the approach shown in our study. Then, to improve the production of prodigiosin in strain JNB5-1, the genes *pigN* and *pigF* were overexpressed under the control of the promoters P_pig_, P_RplJ_, and P_SM_, and strains SM01, SM02, and SM03 were obtained, respectively. The results showed that the prodigiosin titer of these strain was positively correlated with the strength of the promoters. Among them, the prodigiosin titer of strain SM03 increased to 8.52 g/L, which was 1.35-times greater than that of the original strain JNB5-1 (6.32 g/L; Figure 5). This result further suggests that random genomic interruption and FACS technology is an effective method to identify strong constitutive promoter in the host that we are interested in.

## 5. Conclusions

This work describes a novel method to identify strong constitutive promoters in *E. coli* and *S. marcescens* based on random genomic interruption and FACS technology, and the identified promoters were further used in fine-tuning gene expression and reprogramming metabolic flux for higher-level production of L-valine and prodigiosin in *E. coli* and *S. marcescens*, respectively. The method shown in our study can also be a useful strategy for isolating other effective genetic regulatory elements, such as ribosome binding sites, terminators, and N-terminal coding sequences (NCS), in different microorganisms.

## Figures and Tables

**Figure 1 biology-12-00071-f001:**
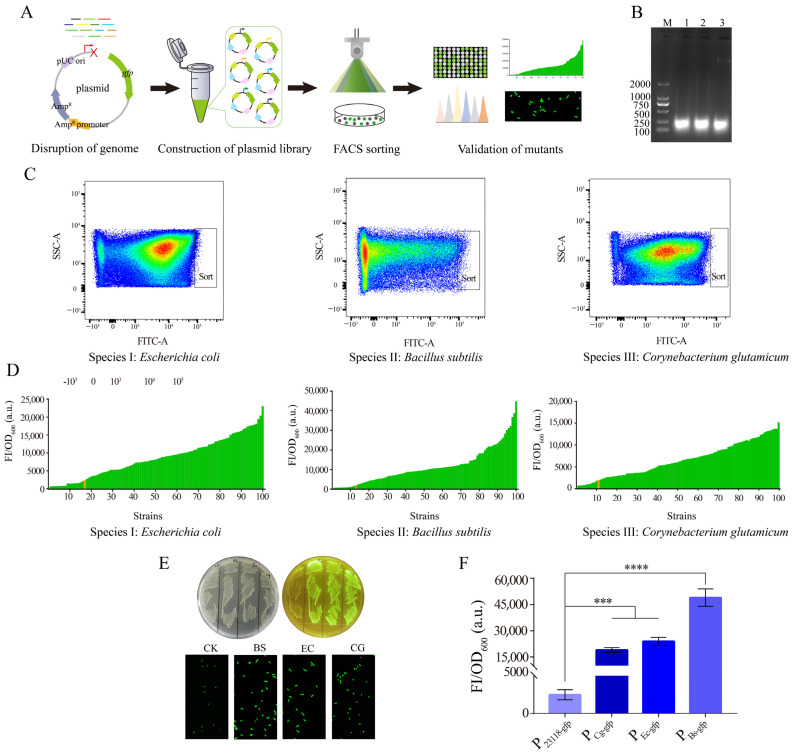
Design and construction of a novel approach for screening strong constitutive promoters. (**A**) Schematic diagram of screening process of the strong constitutive promoters. (**B**) Gel of randomly interrupted genomes of *E. coli* MG1655, *B. subtilis* 168, and *C. glutamicum* ATCC13032. M: Marker DL2000. Lane 1: Random interruption of the *E. coli* MG1655 genome. Lane 2: Random interruption of the *B. subtilis* 168 genome. Lane 3: Random interruption of the *C. glutamicum* ATCC13032 genome. (**C**) FACS sorting of the strong promoters from the three different sources. (**D**) Rescreening of the mutants with higher fluorescence intensities in 96-deep-well plates. The left panel shows the constitutive promoters screened from *E. coli* MG1655, the middle panel exhibits the constitutive promoters screened from *B. subtilis* 168, and the right panel represents the constitutive promoters screened from *C. glutamicum* ATCC13032. The orange band represents the control strain. (**E**) Visual comparison of promoter intensities from three different sources under blue light (upper panel). Fluorescence microscopy images of strains carrying three sources of promoters (lower panel). (**F**) Comparison of fluorescence intensity of the strongest promoters screened from the three sources. For (**F**), error bars indicate standard deviations. One-way analysis of variance (ANOVA) was used to examine the mean differences between the data groups. **** *p* < 0.001, *** *p* < 0.005.

**Figure 2 biology-12-00071-f002:**
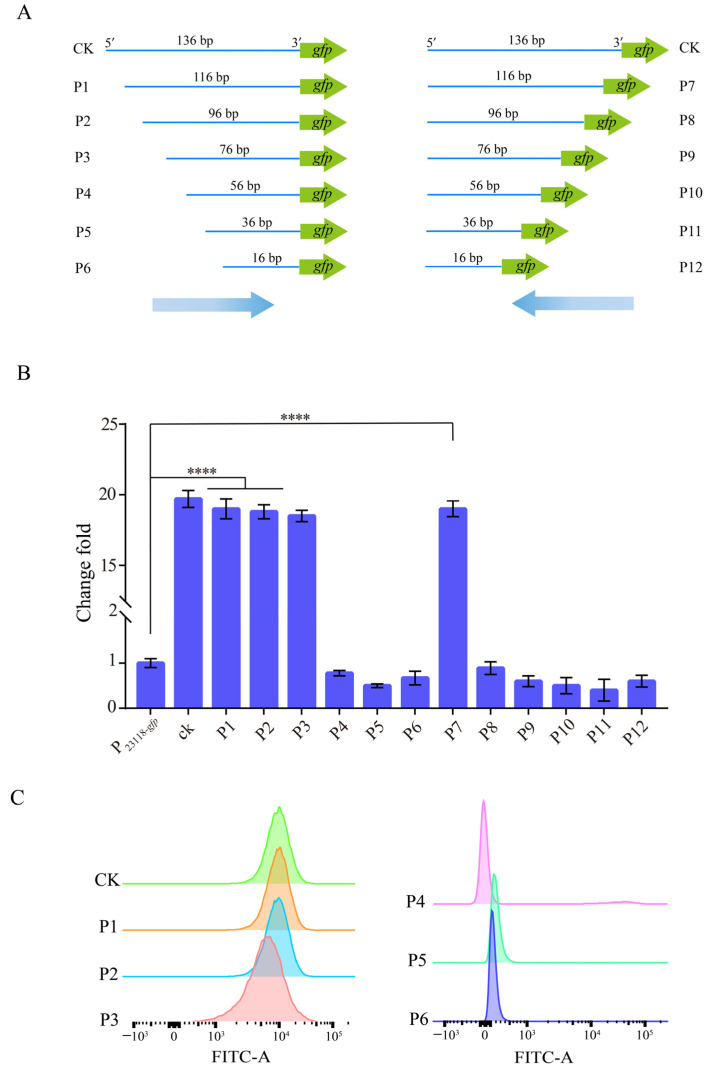
Characterization and identification of the constitutive promoters from *B. subtilis* 168. (**A**) Shortening and characterization of the screened promoter sequence from *B. subtilis* 168. (**B**) Fluorescence variation range of each truncated promoter using GFP as the reporter gene. (**C**) FACS analysis of fluorescence intensity of truncated promoters P1 to P6. For (**B**), error bars indicate standard deviations. One-way analysis of variance (ANOVA) was used to examine the mean differences between the data groups. **** *p* < 0.001.

**Figure 3 biology-12-00071-f003:**
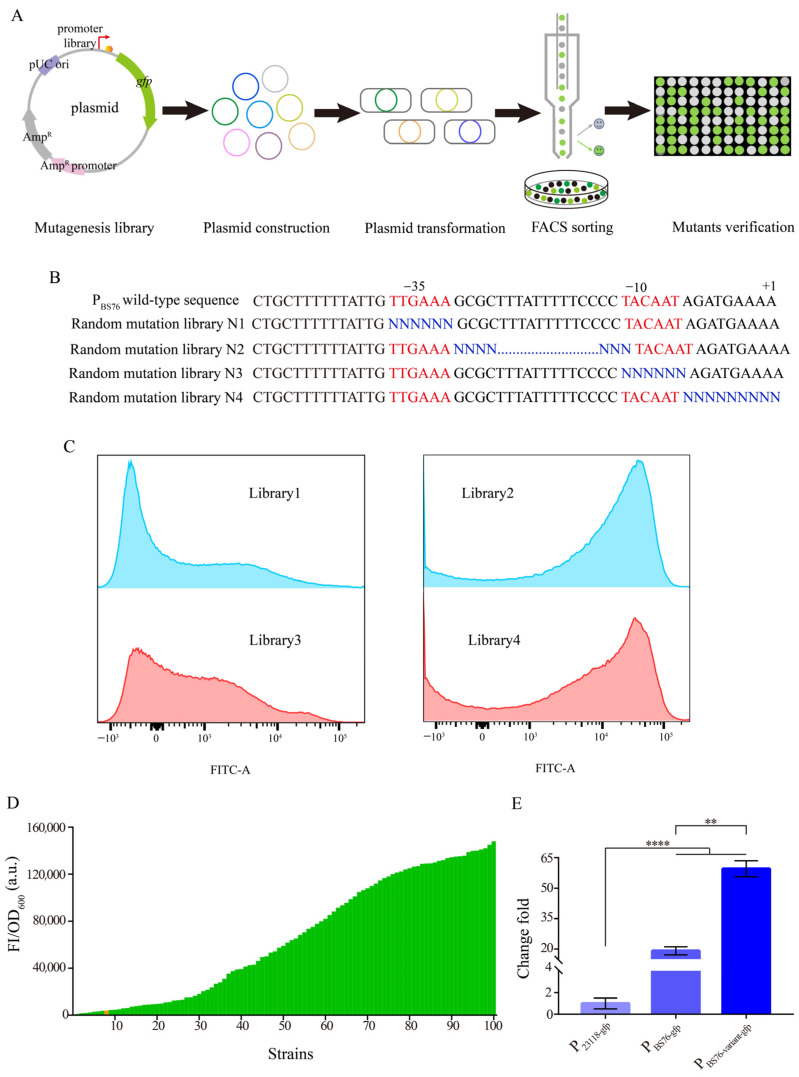
Construction of a gradient promoter library based on the promoter random mutation library. (**A**) Workflow of construction and screening of the promoter random mutation library. (**B**) Specific mutation patterns of the four mutation libraries. The degenerate bases introduced during PCR primer synthesis are highlighted in blue color. The –35 and –10 regions of promoter P_BS76_ are highlighted in red color. The other sequences of promoter P_BS76_ are showed in black color. (**C**) Range of fluorescence intensity of the four constructed libraries. (**D**) Fluorescence intensity of the random selected mutants for the second round of screening. The orange band represents the fluorescence intensity of P_BBa_J23118_-*gfp*. (**E**) Comparison of fluorescence intensity among P_23118_-*gfp*, P_BS76_-*gfp*, and P_BS76-variant_-*gfp*. P_23118-gfp_ indicates P_BBa_J23118-gfp_. For (**E**), error bars indicate standard deviations. One-way analysis of variance (ANOVA) was used to examine the mean differences between the data groups. **** *p* < 0.001, ** *p* < 0.01.

**Figure 4 biology-12-00071-f004:**
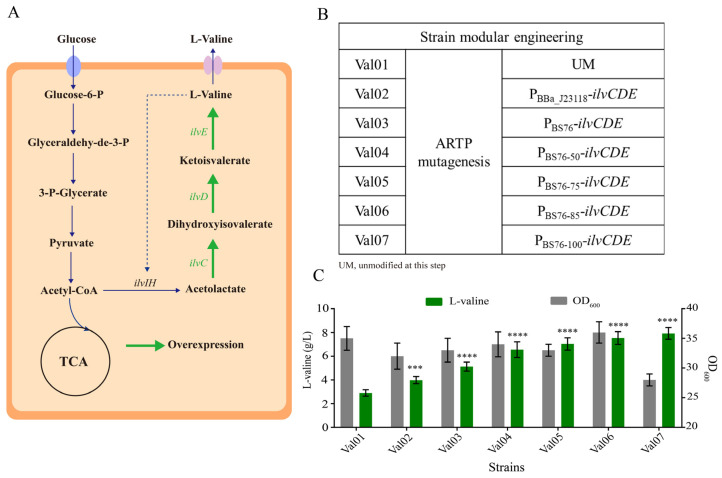
Optimization of the L-valine pathway metabolic flux using the screened gradient promoter in *E. coli* W3110. (**A**) L-valine synthesis pathway in *E. coli* W3110. *ilvIH*, acetolactate synthase; *ilvC*, acetohydroxy acid isomeroreductase; *ilvD*, dihydroxy acid dehydratase; *ilvE*, branched-chain amino acid aminotransferase; dotted line indicates feedback inhibition. (**B**) Enhancing the expression of *ilvCDE* genes by using the gradient strength promoters. Strains Val02 to Val07 were constructed by overexpressing the *ilvCDE* genes with promoters P_BBa_J23118_, P_BS76_, P_BS76-50_, P_BS76-75_, P_BS76-85_, and P_BS76-100_ in strain Val01. (**C**) L-valine titers of strains Val01 to Val07 were detected in shake-flask fermentation. For (**C**), error bars indicate standard deviations. One-way analysis of variance (ANOVA) was used to examine the mean differences between the data groups. **** *p* < 0.001, *** *p* < 0.005.

**Figure 5 biology-12-00071-f005:**
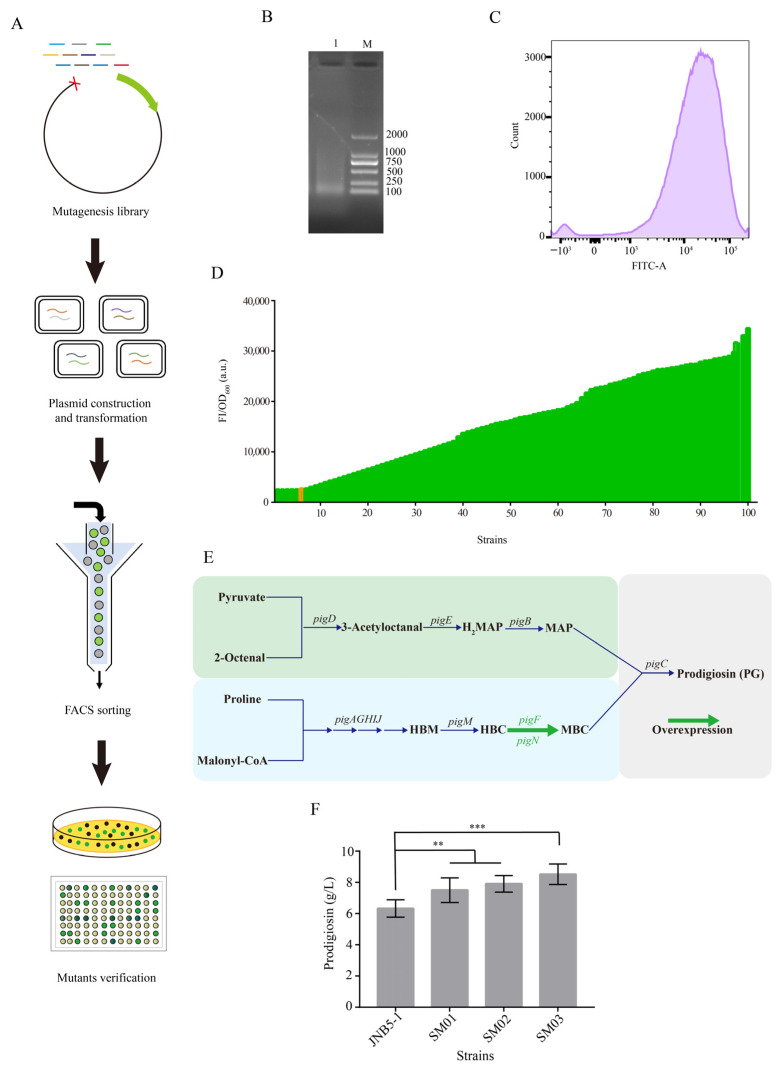
Optimization of prodigiosin synthesis using the strong constitutive promoter screened from *S. marcescens*. (**A**) Workflow of construction and screening of the constitutive promoter. (**B**) Gels of randomly interrupted genomes of *S. marcescens* JNB5-1. M: Marker DL2000. Lane 1: Random interruption of the *S. marcescens* JNB5-1 genome. (**C**) FACS sorting of promoters with enhanced fluorescence intensities from *S. marcescens* JNB5-1. (**D**) Strength analysis of the P_SM_ variants using the *gfp* gene as the reporter. The orange band represents the fluorescence intensity of P_pig_-*gfp*. (**E**) Prodigiosin synthesis pathway in *S. marcescens* JNB5-1. HBM: 4-hydroxy-2,2′-bipyrrole-5-methanol; HBC: 4-hydroxy-2,2′-bipyrrole-5-carbaldehyde; MBC: 4-methoxy-2,2′-bipyrrole-5-carbaldehyde; H_2_MAP: dihydro-2-methyl-3-n-amyl-pyrrole; MAP: 2-methyl-3-n-amyl-pyrrole. (**F**) Prodigiosin titers of strains JNB5-1, SM01, SM02, and SM03 were detected in shake-flask fermentation. For (**F**), error bars indicate standard deviations. One-way analysis of variance (ANOVA) was used to examine the mean differences between the data groups. *** *p* < 0.005, ** *p* < 0.01.

## Data Availability

The data that support the findings of this study are available from the corresponding author upon reasonable request.

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
