# Peer review of "A Novel Method to Screen Strong Constitutive Promoters in Escherichia coli and Serratia marcescens for Industrial Applications"

_biology, 2022, doi:10.3390/biology12010071_

Round 1

Reviewer 1 Report

This manuscript conducted by Pan and co-workers focused on the design and construction of novel methods to isolate strong constitutive promoters in microorganisms. All contents presented here were well, especially, the results is very interesting. however, some minor issues need to be discussed with the authors.

- The genome of E. coli was broken by ultrasonic method, have tried to disrupt it by enzyme digestion?

- Flow cytometry was used to select mutants with high fluorescence intensity from the promoter library. What was the control for the selection?

- Page 1, line 23, “a useful strategy” should change to “an useful strategy”.

- Page 2, line 81 and 82, Escherichia coli, Corynebacterium glutamicum and Bacillus subtilis should be abbreviated.

- Page 3, line 127, Serratia marcescens should be abbreviated.

- Page 5, line 183, should change “pUC18-gfp” to “pUC19-gfp”.

Author Response

Point 1: The genome of E. coli was broken by ultrasonic method, have tried to disrupt it by enzyme digestion?

Response 1: Thank you for your valuable advice, we have tried to disrupt the genome of E. coli by enzyme digestion. However, the gel verification found that there was no required target band, and the genome of E. coli cannot be interrupted by enzyme digestion. The reason might be the presence of the restrictive modification system in E. coli that protects cells from be broken by enzymes or unwanted DNA brought into cells. Thus, we chose to break the genome of E. coli by ultrasonic method.

Point 2: Flow cytometry was used to select mutants with high fluorescence intensity from the promoter library. What was the control for the selection?

Response 2: Thank you for your good question, and we apologize for not elaborating on this issue you mentioned above. First, the three constructed random promoter libraries Library1-EC, Library2-BS and Library3-CG were subjected to FACS sorting for screening the strong constitutive promoters. E. coli/pUC19-PBBa_J23118-gfp was used as the control strain to screen mutants with significantly higher fluorescence intensity. (Please see Page 6, Line 279). Then the screened promoter PBS76 was further optimized by constructing the random mutagenesis library. Here, E. coli/pUC19-PBBa_J23118-gfp and E. coli/pUC19-PBS76-gfp was used as the control strain to screen mutants with significantly higher fluorescence intensity.

Point 3: Page 1, line 23, “a useful strategy” should change to “an useful strategy”.

Response 3: Thank you for your suggestion. As suggested, we have changed the “a useful strategy” to “an useful strategy”. (Please see Page 1, Line 19; Page 3, Line 115; and Page 16, Line 629)

Point 4: Page 2, line 81 and 82, Escherichia coli, Corynebacterium glutamicum and Bacillus subtilis should be abbreviated.

Response 4: As suggested, we have corrected these mistakes. Thank you for your suggestion. (Please see Page 2, Line 71-72)

Point 5: Page 3, line 127, Serratia marcescens should be abbreviated.

Response 5: As suggested, we have corrected this mistake. Thank you for your suggestion. (Please see Page 3, Line 113)

Point 6: Page 5, line 183, should change “pUC18-gfp” to “pUC19-gfp”.

Response 6: Thank you for your suggestion. As suggested, we have changed the “pUC18-gfp” to “pUC19-gfp”. (Please see Page 4, Line 163-164)

Reviewer 2 Report

Authors have generated libraries of E. coli, B. subtilis, and S. marcescens promoters to help engineer biosynthesis of high value compounds. They identified constitutive promoters that enabled them to demonstrate substantive increases in L-valine production from E. coli and prodigiosin production from S. marcescens.

Libraries of promoters are useful for our bioengineering toolkits. However, 1. they should only be a component of more nuanced pathways engineering and 2. Significant resources already exist for E coli and B subtilis promoters. While the promoter library for S. marcescens may be novel, we do not think this warrants publication in Biology.

·      Figures 1: all too small.

·      Describe logic of library construction: random genomic regions are identified as promoters if they drive expression of fluorescent GFP.

·      “randomly interrupted genomes” suggests TnSeq is being used, but fragments are

·      IS this a novel technique?

·      Is the work in E. coli redundant in light of many promoter resources available already? A quick google turns up The E. coli Promoter collection – a library of 1900 promoters driving gfp on low-copy number plasmids.

·      Another quick search shows that there has also been extensive work studying and engineering B. subtilis promoters.

o   E.g. https://journals.plos.org/plosone/article?id=10.1371/journal.pone.0158447

·      As at least one study on the acoR promoter has been published, could this have been identified without using the library approach?

·      Fig 3E – report error in legend and statistical significance

·      This publication [https://pubmed.ncbi.nlm.nih.gov/32961297/] from 2020 reports a yield for L-valine of 84 g/L, 10x greater than the best yields reported here.

·      The increase in prodigiosin production by S. marcescens was very modest – are the yields in fig 5f statistically significant? What are the errors?

·      Discussion does not adequately acknowledge other work on promoter or pathway engineering

Author Response

Point 1: Libraries of promoters are useful for our bioengineering toolkits. However, 1. they should only be a component of more nuanced pathways engineering and 2. Significant resources already exist for E. coli and B. subtilis promoters. While the promoter library for S. marcescens may be novel, we do not think this warrants publication in Biology.

Response 1: Thank you for your suggestion. Although there has been extensive work studying and engineering E. coli and B. subtilis promoters before, the methods to screen strong constitutive promoters in E. coli and B. subtilis were still limited. In the past, promoter-reporter gene assays have been a standard approach for screening strong constitutive promoters and measuring promoter activities. The commonly used reporter genes are the genes encoding chloramphenicol acetyl transferase (cat), β-galactosidase (lacZ), aminoglycoside phosphotransferase (aph), α-amylase (amy), green fluorescent protein (gfp) and red fluorescent protein (mCherry). And techniques to analyze expression profile such as transcriptomics and proteomics have been used for efficient strong constitutive promoter selection methods on genome-wide scale. Among them, due to RNA-sequencing (RNA-seq) could enable the determination of an enormous number of the transcriptional start points and consequently the localization of the respective promoters as well as the promoter activity. Hence, RNA-Seq technology is now the most used method to screen potential strong constitutive promoters. As far as we known, our study is the first study to identify strong constitutive promoters by the method of random genomic interruption and FACS technology. Our method has the advantage of screening of strong constitutive promoters suitable for different microorganisms without the need to determine the omic data, which will save costs to some extent. Also, our approach has the advantage of high throughput. It is worth mentioning that our method can even find some previously undiscovered strong constitutive promoters from different bacterial.

    Also, we apologize for our negligence, because our presentation has brought confusion to the reviewer. In fact, the purpose of constructing higher-level L-valine and prodigiosin producing strains in our study is to further validate the promoter activities identified in our study, and to confirm the effectiveness of our method to screen strong constitutive promoters based on random genomic interruption and FACS technology. Of course, we acknowledge that there is a certain gap between the best yields of L-valine showed in our study and the highest yield reported in the literature, which we will further complete in future research. We hope this is reasonable.

Point 2: Figures 1: all too small.

Response 2: As suggested, we have modified Figure 1. Thank you for your invaluable suggestion. (Please see Page 7, Figure 1)

Point 3: Describe logic of library construction: random genomic regions are identified as promoters if they drive expression of fluorescent GFP.

Response 3: Thank you for your invaluable advice. As suggested, to make it more logical for the method to identify strong constitutive promoters based on random genomic interruption and FACS technology in our study, we have added the sentence “Then, plasmids pUC19-PEC-gfp, pUC19-PBS-gfp and pUC19-PCG-gfp were transformed into E. coli JM109 to isolate strong constitutive promoters in E. coli as the expression of fluorescent GFP are driven by the random genomic regions” in the revised manuscript. (Please see Page 4, Line 164-166)

Point 4: “randomly interrupted genomes” suggests TnSeq is being used, but fragments are

Response 4: As described in the nice paper published by the authors of Tim van Opijnen et al., in the Journal of Nature Methods [1]. Tn-seq is based on the assembly of a saturated Mariner transposon insertion library. After library selection, changes in frequency of each insertion mutant are determined by sequencing the flanking regions en masse. In our study, a novel method based on random genomic interruption and fluorescence-activated cell sorting (FACS) technology was designed to identify strong constitutive promoters in E. coli and S. marcescens for industrial applications. The process of the random genomic interruption contains: (1) The genomes of E. coli MG1655, B. subtilis 168, C. glutamicum ATCC13032 and S. marcescens JNB5-1 were extracted using TIANamp Bacteria DNA Kit. (2) Approaches ultrasound and enzyme digestion were performed to randomly interrupt genomes. In our study, we did not use transposons, nor did we use high-throughput sequencing. Hence, the method to constructed randomly interrupted genomes in our study is not Tn-Seq. However, we still appreciate your invaluable suggestions.

1. Tim van Opijnen, Kip L Bodi, Andrew Camilli. Tn-seq: high-throughput parallel sequencing for fitness and genetic interaction studies in microorganisms. Nat Methods. 2009 Oct; 6 (10): 767-72.

Point 5: IS this a novel technique?

Response 5: The selection of appropriate promoters is a critical factor in promoter engineering, and various approaches have been developed to identified strong constitutive promoter in different microorganisms. Promoter-reporter gene assays have been a standard approach for screening strong constitutive promoters and measuring promoter activities. The commonly used reporter genes are the genes encoding chloramphenicol acetyl transferase (cat), β-galactosidase (lacZ), aminoglycoside phosphotransferase (aph), α-amylase (amy), green fluorescent protein (gfp) and red fluorescent protein (mCherry). And techniques to analyze expression profile such as transcriptomics and proteomics have been used for efficient strong constitutive promoter selection methods on genome-wide scale. Among them, due to RNA-sequencing (RNA-seq) could enable the determination of an enormous number of the transcriptional start points and consequently the localization of the respective promoters as well as the promoter activity. Hence, RNA-Seq technology is now the most used method to screen potential strong constitutive promoters [2-6]. As far as we know, our study is the first study to identify strong constitutive promoters by the method of random genomic interruption and FACS technology. Our method has the advantage of screening of strong constitutive promoters suitable for different microorganisms without the need to determine the omic data, which will save costs to some extent. Also, our approach has the advantage of high throughput.

2. Fanqiang Meng, Xiaoyu Zhu, Ting Nie, Fengxia Lu, Xiaomei Bie, Yingjian Lu, Frances Trouth, Zhaoxin Lu. Enhanced Expression of Pullulanase in Bacillus subtilis by New Strong Promoters Mined From Transcriptome Data, Both Alone and in Combination. Front Microbiol. 2018 Nov 2; 9: 2635.

3. Fengjie Zhao, Xiangsheng Liu, Annie Kong, Yuxin Zhao, Xu Fan, Ting Ma, Weixia Gao, Shufang Wang, Chao Yang. Screening of endogenous strong promoters for enhanced production of medium-chain-length polyhydroxyalkanoates in Pseudomonas mendocina NK-01. Sci Rep. 2019 Feb 12; 9 (1): 1798.

4. Tong Shi, Lu Zhang, Mindong Liang, Weishan Wang, Kefeng Wang, Yue Jiang, Jing Liu, Xinwei He, Zhiheng Yang, Haihong Chen, Chuan Li, Dongyuan Lv, Liming Zhou, Biqin Chen, Dan Li, Li-Xin Zhang, Gao-Yi Tan. Screening and engineering of high-activity promoter elements through transcriptomics and red fluorescent protein visualization in Rhodobacter sphaeroides. Synth Syst Biotechnol. 2021 Sep 30; 6 (4): 335-342.

5. Yunzi Luo, Lu Zhang, Katherine W Barton, Huimin Zhao. Systematic Identification of a Panel of Strong Constitutive Promoters from Streptomyces albus. ACS Synth Biol. 2015 Sep 18; 4 (9): 1001-10.

6. Yingshuo Hou, Siyu Chen, Jianjun Wang, Guizhen Liu, Sheng Wu, Yong Tao. Isolating promoters from Corynebacterium ammoniagenes ATCC 6871 and application in CoA synthesis. BMC Biotechnol. 2019 Nov 12; 19 (1): 76.

Point 6: Is the work in E. coli redundant in light of many promoter resources available already? A quick google turns up The E. coli Promoter collection – a library of 1900 promoters driving gfp on low-copy number plasmids.

Response 6: Thank you for your valuable suggestion. After carefully reading the nice paper published by the authors of Zaslaver et al., in the Journal of Nature Methods [7], we have identified that Zaslaver et al. have not used the method designed in our manuscript to screen strong constitutive promoters suitable for E. coli. Meanwhile, although there are many promoter resources in E. coli, we think this part of work is not redundant. The reason is that our research is mainly to introduce a technology based on genome random interruption and FACS technology to isolate strong constitutive promoters suitable for different microorganisms by high-throughput screening. And as a model strain, E. coli has the advantages of clear genetic background and convenient genetic operation. Therefore, E. coli was selected as the first target in our study to screen strong constitutive promoters using the method we designed. In order to verify the universality of this method in screening of strong constitutive promoters, we subsequently chose the second target, S. marcescens. Based on our results, we believe that the method showed in our study can be also an useful strategy to identified strong constitutive promoters in other bacteria and isolated other effective genetic regulatory elements, such as ribosome binding sites, terminators, and N-terminal coding sequences (NCS) for tuning gene expression in different microorganisms.

7. Alon Zaslaver, Anat Bren, Michal Ronen, Shalev Itzkovitz, Ilya Kikoin, Seagull Shavit, Wolfram Liebermeister, Michael G Surette, Uri Alon. A comprehensive library of fluorescent transcriptional reporters for Escherichia coli. Nat Methods. 2006 Aug; 3 (8): 623-8.

Point 7: Another quick search shows that there has also been extensive work studying and engineering B. subtilis promoters.

E.g. https://journals.plos.org/plosone/article?id=10.1371/journal.pone.0158447

Response 7: Thank you for sharing. We have carefully read the nice paper published by the authors of Song et al., in the Journal of Plos One [8], and we have cited this nice paper in our manuscript (Please see Page 18, Line 742-744). And as we have mentioned above, although there has also been extensive work studying and engineering B. subtilis promoters, the methods to screen strong constitutive promoters in B. subtilis were still limited. In our study, a novel method based on random genomic interruption and fluorescence-activated cell sorting (FACS) technology was designed to screen strong constitutive promoters in E. coli and S. marcescens. Based on our results, we believe that the method shows in our study can be also an useful strategy to identified strong constitutive promoters in B. subtilis. Our method has the advantage of screening of strong constitutive promoters suitable for different microorganisms without the need to determine the omic data, which will save costs to some extent. Also, our approach has the advantage of high throughput. It is worth mentioning that our method can even find some previously undiscovered strong constitutive promoters from different bacteria.

8. Yafeng Song, Jonas M Nikoloff, Gang Fu, Jingqi Chen, Qinggang Li, Nengzhong Xie, Ping Zheng, Jibin Sun, Dawei Zhang. Promoter Screening from Bacillus subtilis in Various Conditions Hunting for Synthetic Biology and Industrial Applications. PLoS One. 2016 Jul 5; 11 (7): e0158447.

Point 8: As at least one study on the acoR promoter has been published, could this have been identified without using the library approach?

Response 8: As mentioned by the reviewer, at least one study on the acoR promoter has been published [9, 10]. Hence, it can be identified without using library approach. However, since the function of AcoR identified in B. subtilis is the regulator of acetoin synthesis, and the expression profiles of acoR maybe influenced by a highly dynamic complex regulatory network. Hence, before our research, we are unlikely to consider that the promoter of acoR gene form B. subtilis can be used as a strong constitutive promoter in E. coli. This result further indicated the method showed in our study can be also an useful strategy to identified strong constitutive promoters suitable for different bacteria, even from other species.

9. N O Ali 1, J Bignon, G Rapoport, M Debarbouille. Regulation of the acetoin catabolic pathway is controlled by sigma L in Bacillus subtilis. J Bacteriol. 2001 Apr; 183 (8): 2497-504.

10. N Krüger, A Steinbüchel. Identification of acoR, a regulatory gene for the expression of genes essential for acetoin catabolism in Alcaligenes eutrophus H16. J Bacteriol. 1992 Jul; 174 (13): 4391-400.

Point 9: Fig 3E – report error in legend and statistical significance

Response 9: Thank you for your good suggestion. As suggested, one-way ANOVA was used for comparing statistical difference between the groups of experimental data in our study. And in our revised manuscript, we have marked the significant differences between the groups of experimental data in each figure. (Please Page 7, figure 1; Page 8, figure 2; Page 10, figure 3; Page 11, figure 4; and Page 13, figure 5)

Point 10: This publication [https://pubmed.ncbi.nlm.nih.gov/32961297/] from 2020 reports a yield for L-valine of 84 g/L, 10x greater than the best yields reported here.

Response 10: Thank you for sharing. We have carefully read the nice paper published by the authors of Hao et al., in the Journal of Metabolic Engineering [11], and we have cited this nice paper in our manuscript (Please see Page 20, Line 799-801). At the same time, we apologize for our negligence, because our presentation has brought confusion to the reviewer. In fact, the purpose of constructing higher-level L-valine producing strains in our study is to further validate the promoter activities identified in our study, and to confirm the effectiveness of our method to screen strong constitutive promoters based on random genomic interruption and FACS technology. Of course, we acknowledge that there is a certain gap between the best yields of L-valine shows in our study and the highest yield reported in the literature, which we will further complete in future research. We hope this is reasonable.

11. Yanan Hao, Qian Ma, Xiaoqian Liu, Xiaoguang Fan, Jiaxuan Men, Heyun Wu, Shuai Jiang, Daoguang Tian, Bo Xiong, Xixian Xie. High-yield production of L-valine in engineered Escherichia coli by a novel two-stage fermentation. Metab Eng. 2020 Nov; 62: 198-206.

Point 11: The increase in prodigiosin production by S. marcescens was very modest – are the yields in fig 5f statistically significant? What are the errors?

Response 11: Thank you for your good suggestion. As suggested, one-way ANOVA was used for comparing statistical difference between the data of prodigiosin production by strains JNB5-1, SM01, SM02, and SM03, and results showed that the prodigiosin production in strains SM01, SM02, and SM03 was significantly higher than that of strain JNB5-1. Also, in our revised manuscript, we have marked the significant differences between the groups of experimental data in figure 5f. (Please see Page 13, figure 5f)

Point 12: Discussion does not adequately acknowledge other work on promoter or pathway engineering

Response 12: Thank you for your invaluable advice. As suggested, to improve the quality of our manuscript, we have carefully re-written the discussion section of our manuscript. (Please see Page 14, 15 and 16)

Reviewer 3 Report

The study by Pan et al. describes a novel approach to identify strong constitutive promoters in Escherichia coli and Serratia marcescens based on random genomic interruption and fluorescence-activated cell sorting (FACS) technology. The identified promoters were further used for fine-tuning gene expression and reprogramming metabolic flux of L-valine and prodigiosin in E. coli and S. marcescens, respectively, and finally a high-level L-valine synthesis strain val05 and prodigiosin production strain SM03 was isolated. This is an interesting and valuable study that will be of interest to a broad audience. I have only some minor questions/comments.

- In lines 406-407, Fluorescence unit (a.u.) should be added after the fluorescence value. Please check the fluorescence value of the full text.

- What does the dotted line in Figure 4A mean? It is not mentioned in the caption and text, can it be removed?

- No control strains are marked in Figures 3D and 5D.

- Is the strongest promoter mentioned in lines 521-522 the same promoter as the PSM promoter in lines 535?

- Which strain is mentioned in lines 547-548? Is it SM03 in Figure 5F? Please add corresponding strains in the text.

- The Escherichia coli, Corynebacterium glutamicum, and Bacillus subtilis in Line 81-82 should be abbreviated.

Author Response

Point 1: In lines 406-407, Fluorescence unit (a.u.) should be added after the fluorescence value. Please check the fluorescence value of the full text.

Response 1: Thank you for your good suggestion. As suggested, we have added the fluorescence unit (a.u.) after the fluorescence value in the revised manuscript. (Please see Page 9, Line 374, 375, and 378; Page 12, Line 474 and 475)

Point 2: What does the dotted line in Figure 4A mean? It is not mentioned in the caption and text, can it be removed?

Response 2: The Dotted line shows in Figure 4A indicate feedback inhibition. As suggested, we have added the sentence “Dotted line indicates feedback inhibition” in the revised manuscript. Thank you for your suggestion. (Please see Page 11, Line 441-442)

Point 3: No control strains are marked in Figures 3D and 5D.

Response 3: Thank you for your good suggestion. As suggested, we have marked the control strains in figures 3D and 5D. (Please see Page 10, Figure 3D; and Page 13, Figure 5D)

Point 4: Is the strongest promoter mentioned in lines 521-522 the same promoter as the PSM promoter in lines 535?

Response 4: Sorry, due to our negligence, we failed to give a more detailed description of this part. In fact, the strongest promoter mentioned in lines 521-522 is the same promoter as the promoter PSM in line 535. In the revised manuscript, to explain the strongest promoter mentioned in lines 521-522, we have changed the sentence from “The fluorescence intensity of one of the mutants was significantly higher than the others” to “The fluorescence intensity of one of the mutants that contain the gfp gene under the control of the promoter PSM (the promoter of the operon rpsQ-rpsJ) was significantly higher than the others”. We hope this is reasonable. (Please see Page 12, Line 479-480)

Point 5: Which strain is mentioned in lines 547-548? Is it SM03 in Figure 5F? Please add corresponding strains in the text.

Response 5: Thank you for your suggestion. As suggested, we have added the corresponding strain SM03 in the revised manuscript. (Please see Page 13, Line 502-503)

Point 6: The Escherichia coli, Corynebacterium glutamicum, and Bacillus subtilis in Line 81-82 should be abbreviated.

Response 6: Thank you for your suggestion. As suggested, we have corrected these mistakes. (Please see Page 2, Line 71-72)